# Prediction of Array Antenna Assembly Accuracy Based on Auto-Encoder and Boosting-OSKELM

**Yifei Tong \*, Miao Wang and Tong Zhou**

School of Mechanical Engineering, Nanjing University of Science & Technology, Nanjing 210094, China
\* Correspondence: tyf1776@mail.njust.edu.cn

**Abstract:** As a critical component for space exploration, navigation, and national defense, array antenna secures an indispensable position in national strategic significance. However, various parts and complex assembly processes make the array antenna hard to meet the assembly standard, which causes repeated rework and delay. To realize the accurate and efficient prediction of the assembly accuracy of array antenna, a prediction method based on an auto-encoder and online sequential kernel extreme learning machine with boosting (Boosting-OSKELM) is proposed in this paper. The method is mainly divided into two steps: Firstly, the auto-encoder with the fine-tuning trick is used for training and representation reduction of the data. Then, the data are taken as the input of Boosting-OSKELM to complete the initial training of the model. When new sample data is generated, Boosting-OSKELM can realize the online correction of the model through rapid iteration. Finally, the test shows that the average MSE of Boosting-OSKELM and ANN is 0.061 and 0.12, and the time consumption is 0.85 s and 15 s, respectively. It means that this method has strong robustness in prediction accuracy and online learning ability, which is conducive to the development of array antenna assembly.

**Keywords:** array antenna; assembly accuracy; auto-encoder; boosting-OSKELM





## 1. Introduction

The electronic information industry is an important driving force for today's economic and social development. It is a strategic, basic, and leading pillar industry of the national economy, which plays an important role in promoting economic growth, industrial structure, changing development mode, and maintaining national security [1,2]. In recent years, high-precision integrated antenna (array antenna), as an indispensable part of early warning and detection systems, has become the core of the national major project "Space-Earth Integration Network" [3]. The basic structure of the array antenna body is shown in Figure 1. It can be mainly divided into three parts, namely the antenna subarray element, the function layer, and power. However, the specific configuration of the array antenna is rather complex. Taking the new exploration satellite as an example, its array antenna is composed of 500 subarrays and more than 1 million parts, with over 100 million assembly welding points. What's more, it needs to be in trouble-free service for more than 8 years in the harsh space environment, which features extremely high requirements.

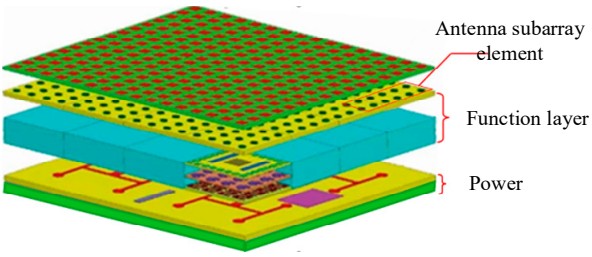

**Figure 1.** The basic structure of array antenna body.

Due to the complexity of the structure and high requirements for quality, array antennas are mostly assembled manually or with the help of mechanical equipment [4]. With the uncertainty of assembly activities, the assembly process is often unqualified or the performance of the mechanical and electrical cannot meet the design standard, which results in an average of more than 12 months of repeated adjustment, seriously reducing the assembly efficiency [5]. Therefore, in order to shorten the assembly cycle of array antennae and guarantee product quality, it is urgent to introduce an effective method of assembly accuracy prediction to the manufacturing of array antennae.

There have been many related researches in this field. In order to study the effect of assembly error on the antenna gain, Guo et al. [6] proposed an accurate gain prediction model using an improved XGBoost algorithm and the transfer learning method, based on the simulation data and experience. Combined with the fact that the geometric characteristics of parts/components of the aero-engine rotor are not related to the measurement datum, Liu et al. [7] proposed a datum error elimination method that makes the rotor characteristic matrix and assembly model more accurate, thereby improving the prediction effect of assembly accuracy. Mu et al. [8] studied the construction method of composite processing components considering the manufacturing error and deformation factors of parts and proposed a new prediction method for aero-engine high-pressure rotor systems. Aiming at the goals of Zero-defect Manufacturing, Elisa et al. [9] established a diagnostic tool that provides an in-line identification of critical steps of assembly processes. The methodology is based on a self-adaptive defect prediction model of the process, which can be updated with the input of new data. The research mentioned above either established a data-driven model in regard to historical data and simulation data or built a mechanical model based on physical principles. However, both of them are off-line prediction models, which is difficult to achieve rapid prediction in some complex cases. Moreover, it also lacks the ability of model iteration, leading to a low efficiency in dealing with new sample data for online correction of the model.

In recent years, digital twin technology has been widely studied and applied in advanced manufacturing. As a virtual-physical fusion technology, it can realize the virtual-physical interaction, data fusion, decision analysis, and iterative optimization of the whole assembly process by using the twin data of assembly context, based on the virtual assembly information model and quantitative calculation of assembly quality [10]. Obviously, the implementation of digital twin technology is inseparable from artificial intelligence technology, which is equipped with the ability of high-performance data analysis and real-time prediction. However, the assembly process of an array antenna has the following features, which bring difficulties to assembly accuracy prediction:

(1) The assembly process of an array antenna is complex and variable, and the data of its assembly samples are mostly in high-dimensional space, so a single sample may have redundant or even contradictory information between sample features, which is a disadvantage to the prediction of assembly accuracy.

(2) Array antenna belongs to the small batch production mode, and its historical sample data has limitations, which will disturb the training of the machine learning model.

As to the problem of high-dimension, data mining technology can be used to extract valuable information [11]. The commonly used data mining method is representation reduction, which is a dimension-reducing or feature extraction method. It can reduce the distance between sample points in high-dimensional space and preserve the valuable information of samples as much as possible, which is conducive to improving the inference speed and accuracy of the prediction algorithm [12]. In traditional machine learning, the commonly used feature extraction methods are mainly divided into two parts, that is, linear dimension reduction represented by PCA [13] and nonlinear dimension reduction based on manifold learning. PCA may bring more information loss in nonlinear problems, while manifold learning (such as LLE [14], t-SNE [15], etc.) starts from the relationship between samples, which leads to a high effectiveness of dimension reduction but also a

high computational complexity. So the traditional method cannot calculate the dimension reduction at the second level.

For Few-shot learning problems in engineering projects, the Kriging algorithm is often used. Although this method does not rely on the amount of data, it still depends on the distribution of data. If the distribution of data is poor, the prediction accuracy of Kriging may be greatly affected [16]. In addition, statistical learning methods such as the K-nearest neighbor algorithm (KNN) [17] and Support Vector Regression (SVR) [18] can also adapt to Few-shot learning problems. But these algorithms often use offline learning, and there is no good online incremental learning method for new sample data generated in the future.

In view of the issues above, the main contributions of the study in this paper are as follows:

(1)    An improved auto-encoder is proposed to implement feature extraction, which not only has high computational efficiency but also can adapt to small sample problems.
(2)    An online sequential extreme learning machine with Boosting strategy(Boosting-OSKELM) is proposed to adapt to online learning. This method possesses a fast speed of learning and model iteration, which can meet online learning requirements.

## 2. Methodology

### 2.1. Representation Reduction of Sample Data

2.1.1. Dimension Reduction Principle of Auto-Encoder

In recent years, neural networks have made remarkable achievements in many fields, such as genetics [19], graph classification [20], medical diagnosis [21], fault diagnosis [22], and so on. Based on the general approximation theorem [23], it can fit any function in theory. Its general structure is mainly divided into three parts: input layer, hidden layer, and output layer. The input layer and output layer correspond to the input data and prediction results respectively, and the middle hidden layer introduces a nonlinear activation function, which enables the neural network to learn more hidden feature information. Therefore, the neural network is good at multi-level representation and data prediction of nonlinear systems.

In the field of data dimension reduction, the common model of the neural network is auto-encoder (AE), which is a neural network aiming at restoring the input data to the greatest extent [24]. Figure 2 is the structure of a single-layer auto-encoder (SAE), which is fully connected among layers. Assuming that the sample $i$ is $x(i) \in R^d$, the AE will map this sample to a new feature space $z(i) \in R^m$; then, the AE will reconstruct the new sample representation into the original one, and the reconstructed sample is defined as $x'(i) \in R^d$.

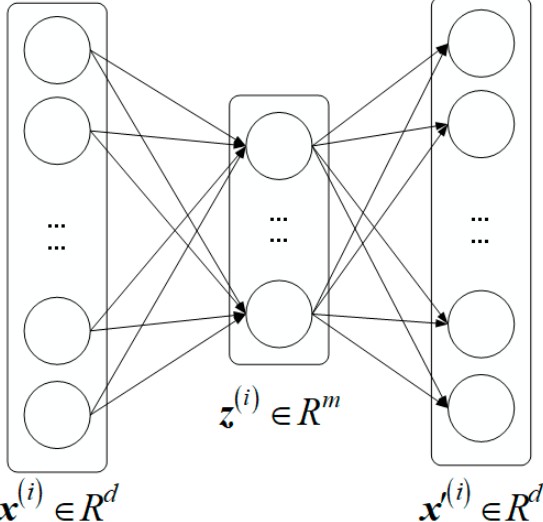

**Figure 2.** Single-layer auto-encoder.

As can be seen from the above figure, the AE can be divided into two parts: encoder $f : R^d \to R^m$ and decoder $g : R^m \to R^d$. The learning goal of the AE is to minimize the reconstruction loss shown in Equation (1).

$$L = \sum_{i=1}^{N} \left\| x^{(i)} - g\left(f\left(x^{(i)}\right)\right) \right\|^2 \tag{1}$$

where $||\cdot||$ is the vector-2 norm, and the mapping relationship $f$ from $x(i)$ to $z(i)$ is represented in Equation (2), where $W^{(1)}$ and $b^{(1)}$ respectively represent the connection weight and bias of the encoder part. Similarly, there is a mapping relationship $g$ from $z(i)$ to $x'(i)$, which is represented by Equation (3).

$$z^{(i)} = f\left(W^{(1)}x^{(i)} + b^{(1)}\right) \tag{2}$$

$$x'^{(i)} = g\left(W^{(2)}z^{(i)} + b^{(2)}\right) \tag{3}$$

If the dimension number of the feature space $m$ is less than the dimension of the original space $d$, the AE can be regarded as a feature extraction method. The following is a sample representation reduction method. If necessary, in order to prevent over-fitting, the binding weight can be considered, that is $W^{(1)} = W^{(2)}$. Then the regularization is used, as shown in Equation (4), where $||\cdot||_F$ represents the $F$ norm of the matrix, $\lambda$ is the coefficient of regularization.

$$L = \sum_{i=1}^{N} \left\| x^{(i)} - g\left(f\left(x^{(i)}\right)\right) \right\|^2 + \lambda \|W\|_F^2 \tag{4}$$

In some scenarios, the number of hidden layers can be increased to form a deep auto-encoder (DAE). Theoretically, deeper layers of neural networks mean more neurons and stronger learning performance. However, in practice, blindly stacking layers may lead to gradient disappearance or gradient explosion, resulting in the non-convergence of neural training. Therefore, the construction of a neural network requires skills and experience.

2.1.2. Fine-Tuning Trick

In early neural network training, if the random initialization strategy is adopted for the model parameters, the training of deep networks will be difficult. In this regard, when training a deep belief network, Hinton et al. [25] proposed a greedy pre-training method, to pre-train the restricted Boltzmann machine of each layer, and finally add the output layer. This method is called fine-tuning, which is a common and important deep-learning training skill. At present, it has been widely used in many fields of artificial intelligence, especially in the field of transfer learning. The basic idea of fine-tuning is that, based on network pre-training, the network is modified through the traditional global learning algorithm to make the model converge to a better local optimum.

This paper uses the fine-tuning trick based on supervised learning. The basic process is shown in Figure 3. Firstly, the decoder output of AE is combined with the input of multi-layer perceptron (MLP), namely the output of the encoder is used as the input of MLP. Then the network structure is used to train the regression problem on the dataset, which is called pre-training. The reason for choosing MLP is that the network weight parameters of AE can be updated through gradient backpropagation. After pre-training, the AE structure is separated and connection weights in the pre-training stage are retained. On this basis, the weight of AE is retrained, which is called tuning. After the fine-tuning is completed, the decoder part of AE is removed, while the encoder is retained, and the representation reduction model of the sample data is obtained.

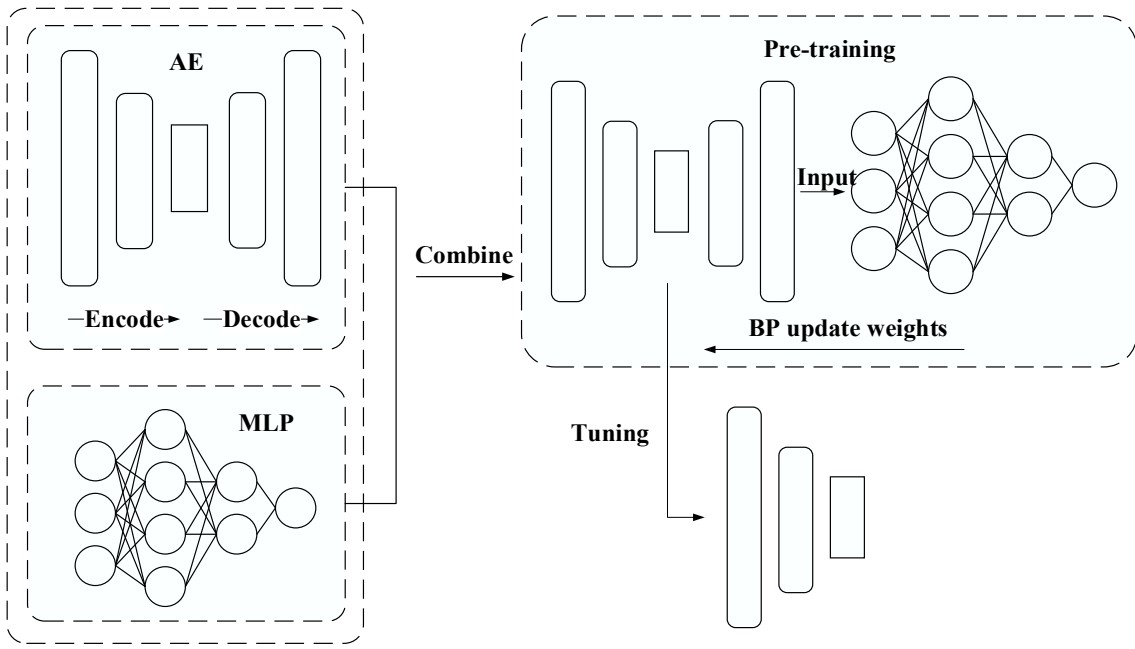

**Figure 3.** The basic process of fine-tuning.

*2.2. Online Learning Model*

In theory, the network structure of the encoder and MLP can be directly used to predict assembly accuracy. However, due to the lack of available training samples, the generalization performance of the trained neural network may be poor. In addition, from the perspective of model online correction, although neural networks can correct the model through the gradient descent method, it heavily relies on batch data. If it is corrected by only a small number of samples, the prediction performance of the model may be sharply reduced due to the problem of "data poison" [26].

In order to solve the problem of model online learning, scholars have proposed a large number of online learning algorithms. Among them, the online sequential extreme learning machine (OSELM) is favored by many scholars because of its fast training speed and good generalization performance, which can achieve online prediction of sample data efficiently [27]. It has the following advantages:

(1) The learning speed of OSELM is very fast, which avoids the disadvantage of slow back-gradient updates of traditional BP neural networks.
(2) It is easy for OSELM to obtain the global optimal solution. The optimization model, namely the least square method, is used to solve the network weight.
(3) OSELM has few parameters, which avoids the great influence of learning rate parameters on the performance of the BP neural network;
(4) As OSKELM is free from the gradient descent method, while updating parameters through matrix transformation, its calculation speed is faster so it has a strong online learning ability. Additionally, because of its simple structure, OSKELM features a lower chance of overfitting, which makes it adapt to small sample data.

OSELM derives from extreme learning machines (ELM) [28]. Based on this, the incremental learning formula of new samples is achieved. ELM is a feedforward neural network, and its basic structure is shown in Figure 4, where $w$ represents the weight between the input layer and hidden layer neurons, $b$ stands for biases, $\beta$ is the weight between the hidden layer and output layer neurons, and $g$ is the activation function of hidden layer neurons.

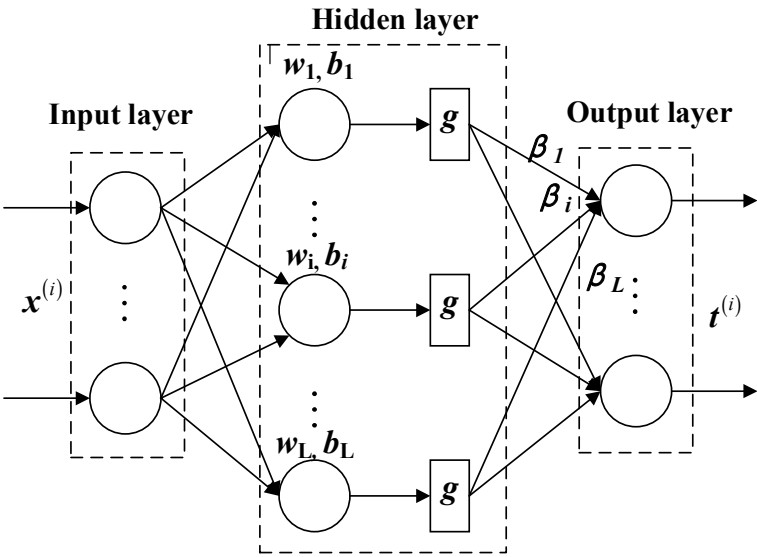

**Figure 4.** Single-layer ELM.

Different from the current popular deep neural network, it does not update the network weight through back-gradient propagation but solves the weight value through Moore–Penrose generalized inverse. Define dataset with n samples $(X, T)$, where $X = [x^{(1)}, x^{(2)}, \ldots, x^{(N)}]^T$, $x^{(n)} \in R^d$, and $T = [t^{(1)}, t^{(2)}, \ldots, t^{(N)}]^T$, $t^{(n)} \in R^m$. If the number of neurons in the hidden layer is $L$ and its activation function is $g$, the final output of ELM on the data set is shown in Equation (5), $w_i \in R^d$, $b_i \in R$.

$$\begin{aligned} f_L(X) &= \sum_{i=1}^{L} \beta_i g_i(w_i X + b_i) \\ &= Y, n = 1, 2, \ldots, N \end{aligned} \tag{5}$$

Define $h_i = g_i(w_i X + b_i)$ and convert it to a matrix representation to obtain the output shown in Equation (6).

$$f_L(X) = H\beta \tag{6}$$

where $H = [h_1(X), \ldots, h_L(X)]_{N \times L}$, $\beta = [\beta_1, \ldots, \beta_L]^T$, $\beta_l \in R^m$.

ELM training is mainly divided into two stages. The first stage is random mapping, where ELM randomly initializes the weight $w_i$ and bias $b_i$ from the input layer to the hidden layer. The second stage is linear parameter solving. According to the weight and bias in the first stage, combined with the optimization problems shown in Equations (6) and (7), then $\beta$ can be solved.

$$\min\|H\beta - T\|^2 \tag{7}$$

In order to enhance the stability of $H$ (the nonsingular matrix), the regularization coefficient $C$ and identity matrix $I$ are introduced. And because the matrix $H$ is often row full rank, so the optimal value $\beta$ is shown in Equation (8), which $H^+$ represents the Moore–Penrose generalized inverse matrix $H$.

$$\beta^* = H^+ T = H^T \left( HH^T + \frac{I}{C} \right)^{-1} T \tag{8}$$

Although the learning speed of ELM is very fast, its prediction performance still lags behind the popular deep neural network. In order to enhance the nonlinear fitting ability of ELM, a kernel function is introduced to form a kernel extreme learning machine (KELM) [29].

The kernel function $K\left(x^{(i)}, x^{(j)}\right) = h\left(x^{(i)}\right) \cdot h\left(x^{(j)}\right)$ is a common method to solve non-linear problems. It maps the data in the original feature space to the new high-dimensional one. Learning is implicit in the new feature space, and there is no need to explicitly define the kernel mapping function in the feature space. The kernel matrix $\Omega = HH^T$ is defined according to Mercer condition [30], as shown in Equation (9).

$$\Omega = HH^{\mathrm{T}} = \begin{bmatrix} K\left(x^{(1)}, x^{(1)}\right) & \cdots & K\left(x^{(1)}, x^{(N)}\right) \\ \cdots & \cdots & \cdots \\ K\left(x^{(N)}, x^{(1)}\right) & \cdots & K\left(x^{(N)}, x^{(N)}\right) \end{bmatrix}_{N \times N} \tag{9}$$

Common kernel functions are as follows:
(1) Polynomial kernel function

$$K\left(x^{(i)}, x^{(j)}\right) = \left(a \cdot x^{(i)} \cdot x^{(j)} + b\right)^{p} \tag{10}$$

(2) Gaussian kernel function

$$K\left(x^{(i)}, x^{(j)}\right) = \exp\left(-\frac{\left\|x^{(i)} - x^{(j)}\right\|^{2}}{2\sigma^{2}}\right) \tag{11}$$

(3) Linear kernel function (i.e., no kernel)

$$K\left(x^{(i)}, x^{(j)}\right) = x^{(i)} \cdot x^{(j)} \tag{12}$$

where $a, b, p, \sigma$ are constants.

It can be clearly seen that the introduction of kernel function makes ELM no longer affected by random weights $w\ b$, and the prediction of the new sample $x$ can be calculated directly according to Equation (13).

$$\begin{aligned} f(x) &= h(x)\beta^{*} \\ &= \left[K\left(x, x^{(1)}\right), \dots, K\left(x, x^{(N)}\right)\right]\left(HH^{\mathrm{T}} + \frac{I}{C}\right)^{-1}T \end{aligned} \tag{13}$$

When the kernel function is introduced into OSELM, an online sequential kernel extreme learning machine (OSKELM) is formed. For the dataset $\left\{\left(x^{(i)}, t^{(i)}\right)\right\}_{i=1}^{t}$, which is set up to time t, the prediction result is $f(x) = h(x)\beta_t$, where $\beta_t$ is $\beta$ at time t.

Define $k_t(x)$ and $\theta_t$, as is shown in Equations (14) and (15), then Equation (16) can be derived.

$$k_t(x) = \left[K\left(x, x^{(1)}\right), K\left(x, x^{(2)}\right), \dots, K\left(x, x^{(t)}\right)\right] \tag{14}$$

$$\theta_t = \left(H_t H_t^{\mathrm{T}} + \frac{I}{C}\right)^{-1}T_t \tag{15}$$

$$\begin{aligned} f(x) &= \left[K\left(x, x^{(1)}\right), \dots, K\left(x, x^{(t)}\right)\right]\left(H_t H_t^{T} + \frac{I}{C}\right)^{-1}T_t \\ &= k_t(x)\,\theta_t \end{aligned} \tag{16}$$

Moreover, define $A_t = H_t H_t^{\mathrm{T}} + C^{-1}I$, then $\theta_t = A_t^{-1}T_t$. For new samples $\left(x^{(t+1)}, t^{(t+1)}\right)$ in time $t + 1$, the matrix $A_{t+1}$ as follows.

$$A_{t+1} = \begin{bmatrix} A_t & \widetilde{k}_t\left(x^{(t+1)}\right) \\ \widetilde{k}_t^{\mathrm{T}}\left(x^{(t+1)}\right) & v_t \end{bmatrix} \tag{17}$$

where

$$\widetilde{k}_t\left(x^{(t+1)}\right) = \left[K\left(x^{(t+1)}, x^{(1)}\right), \ldots, K\left(x^{(t+1)}, x^{(t)}\right)\right]^{\mathrm{T}} \tag{18}$$

$$v_t = C^{-1} + K\left(x^{(t+1)}, x^{(t+1)}\right) \tag{19}$$

The inverse matrix of the matrix $A_{t+1}$ is obtained according to the inverse formula of the block matrix, as shown in Equation (20).

$$A_{t+1}^{-1} = \begin{bmatrix} A_t^{-1} + A_t^{-1}\widetilde{k}_t\left(x^{(t+1)}\right)\rho_t^{-1}\widetilde{k}_t^{\mathrm{T}}\left(x^{(t+1)}\right)A_t^{-1} & -A_t^{-1}\widetilde{k}_t\left(x^{(t+1)}\right)\rho_t^{-1} \\ -\rho_t^{-1}\widetilde{k}_t^{\mathrm{T}}\left(x^{(t+1)}\right)A_t^{-1} & \rho_t^{-1} \end{bmatrix} \tag{20}$$

where $\rho_t = v_t - \widetilde{k}_t^{\mathrm{T}}\left(x^{(t+1)}\right)A_t^{-1}\widetilde{k}_t\left(x^{(t+1)}\right)$. Thus, the iterative formula of the kernel function coefficient vector is obtained, as shown in Equation (21). In this way, the new samples can be predicted according to Equation (16).

$$\theta_{t+1} = A_{t+1}^{-1}T_{t+1} \tag{21}$$

To sum up, the online training of OSKELM does not need to organize the old and new data together for retraining, but to absorb the new sample information by updating the matrix $A_{t+1}$. After updating, the old dataset information will not be needed, which greatly reduces the computational complexity and improves efficiency.

*2.3. Boosting-OSKELM*

The goal of the supervised learning algorithm is to train stable models that perform well in all aspects. However, most of the time, the performance of supervised learning algorithms can only have decent performance in specific fields, which is also called weak learning. According to ensemble learning theory, weak learners and strong learners are actually equivalent, as several weak learners can obtain the same prediction performance as strong learners through a special combination method. Among them, boosting is a common ensemble learning skill. Its basic idea is to correct the wrong prediction of other weak learners through weak learners.

Based on the idea of boosting, in order to squeeze the performance of the learner as much as possible, this paper ensembles multiple limit learners with different kernels, and proposes the Boosting-OSKELM algorithm, which is shown in Figure 5. Firstly, the polynomial kernel OSKELM is used on the original dataset for training. Then, calculate the residual between the fitting result and the real result on the training set, replace the label of the original data with the residual, and then use Gaussian kernel OSKELM to learn and predict the new residual label. Finally, calculate the residual again, namely the 'residual' of the residual, and so on.

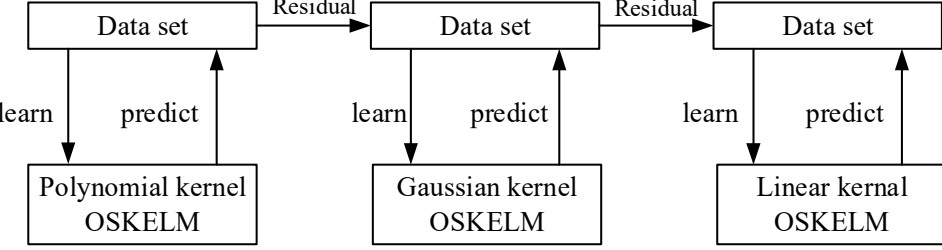

**Figure 5.** Improvement of OSKELM based on Boosting.

The pseudo-code of Boosting-OSKELM is shown in Algorithm 1. In practice, the training of the model does not have to follow the lifting order of the Polynomial kernel → Gaussian kernel → Linear kernel, but can be adjusted appropriately, or even reuse a kernel, such as Gaussian kernel → Gaussian kernel → Gaussian kernel.

| **Algorithm 1:** Boosting-OSKELM |
|---|

**Input:** dataset: $D = \{x,y\}^N$, *kernels* $= \{k_1, k_2, k_3\}$;
**Output**: Ensemble Learning $E^*$;
1.   Init $k = k_1$;
2.    $\hat{y} = \text{OSKELM}(k, x, y)$
3.   **For** $k = \{k_2, k_3\}$:
4.   Calculate residual: $r = y - \hat{y}$;
5.   Fit residual: $\hat{r} = OSKELM(k, x, r)$;
6.   Calculate: $\hat{y} = y + \hat{r}$
7.   **End**
get final Ensemble Learning $E^*$

## 3. Case Study

### 3.1. Data Description

The simplified array antenna subarray unit is shown in Figure 6. It is a stacking structure of three-layer flexible plates, including a soaking plate, PCB plate, and backing plate from bottom to top. The surface of the soaking plate has circular bosses with different heights, which are used to insert different panels. There will be slightly raised bosses on the surface of the PCB for welding the connector, and the other end of the connector is assembled with the insertion pins at the bottom of the backing plate. The connection between the panels is fixed by screws, while different screw preloads make different degrees of deformation between the panels, resulting in greater stress on the welding position of the connector, which seriously affects the assembly quality of the array antenna.

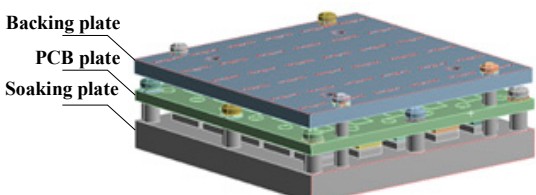

**Figure 6.** Simplified solid model of subarray element.

Based on ANSYS simulation software, deformation simulations under different preload were carried out, one of which is shown in Figure 7. It can be seen that because the backing plate will be arched to the middle under the action of preload, the middle area features large relative displacement, namely the main source of assembly error. In order to measure the assembly accuracy conveniently, this paper selects the relative displacement in the X and Y direction of the position shown in Figure 8 as the prediction target.

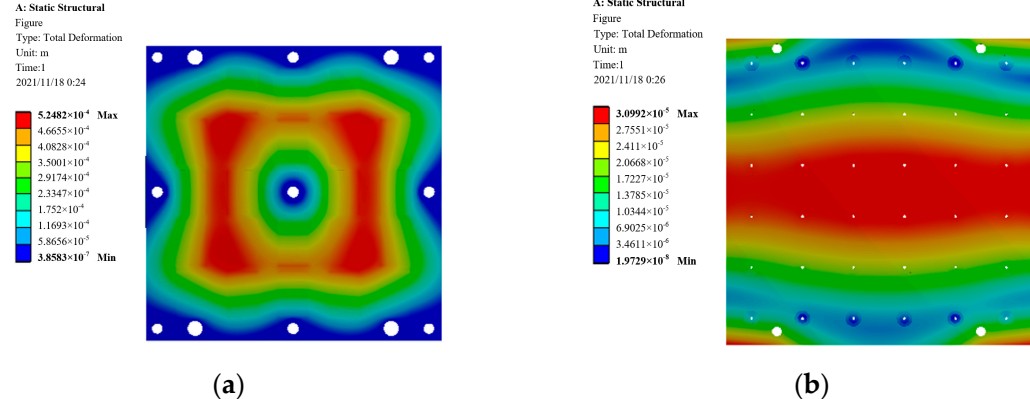

(**a**)                    (**b**)

**Figure 7.** Deformation simulation based on ANSYS (**a**) PCB plate deformation; (**b**) PCB backing plate deformation.

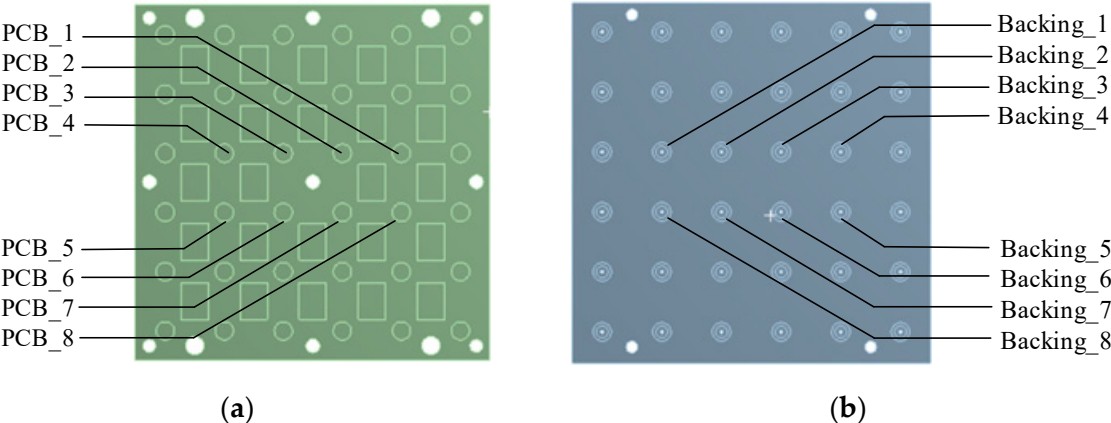

**Figure 8.** Selected assembly accuracy prediction position: (**a**) positions in the PCB plate; (**b**) positions in the backing plate.

In this case, the input data of the model is the preload of each screw (13 screws in total), and the output is the relative displacement of each position shown in Figure 7. This paper uses numerical simulation in ANSYS simulation, and 40 corresponding data are obtained as shown in Tables 1 and 2. According to the simulation results, it was found that the deformation between each sample point is relatively close, which shows that the MinMax normalization strategy is more suitable for data normalization. As a result, the data in both Tables 1 and 2 are normalized according to Equation (22).

$$x' = \frac{x - x_{\min}}{x_{\max} - x_{\min}} \tag{22}$$

**Table 1.** Preload data of 40 screws.

|  | **Screw 1** | **Screw 2** | ... | **Screw 13** |
|---|---|---|---|---|
| Sample 1 | 0.7237 | 0.8845 | ... | 0.904 |
| Sample 2 | 0.7784 | 0.4471 | ... | 0.8941 |
| ... | ... | ... | ... | ... |
| Sample39 | 0.3464 | 0.7638 | ... | 0.4953 |
| Sample40 | 0.6692 | 0.7862 | ... | 0.7256 |

**Table 2.** Displacement data of 40 screws.

|  | $\Delta_{1x}$ | $\Delta_{1y}$ | ... | $\Delta_{8x}$ |
|---|---|---|---|---|
| Sample1 | 0.6809 | 0.6776 | ... | 0.6797 |
| Sample2 | 0.2165 | 0.2235 | ... | 0.2179 |
| ... | ... | ... | ... | ... |
| Sample39 | 0.4779 | 0.4790 | ... | 0.4766 |
| Sample40 | 0.1661 | 0.1617 | ... | 0.1678 |

*3.2. Auto-Encoder Training*

The training of neural networks is different from traditional machine learning. It involves a large number of hyper-parameters, including the network structure, learning rate, and the selection of learners. If using traditional grid search, it will occupy a large amount of computing resources. Therefore, when training the AE, this paper gives empirical fixed values for some hyper-parameters.

(1) As to network architecture, based on the number of input features, the structure of the hidden layer should be as simple as possible. Otherwise, the complex structure will easily cause overfitting. Based on experiences, the initial AE architecture is determined as 13-8-13;

(2)　As to activation functions, the commonly used nonlinear activation functions are the ReLU function, Sigmoid function, and Tanh function. Compared with the other two functions, the Tanh function has a relatively wide output range, which is more conducive to distinguishing the reduced representation between samples;

(3)　As to optimizer, the adaptive moment estimation (Adam) proposed by Kingma et al. [31] is used, which retains the advantages of SGD (stochastic gradient descend) and introduces the momentum, so that the convergence speed of the neural network is accelerated and the learning rate can gradually decline with the number of iterations, which helps find a better local optimal solution. The initial learning rate is 0.01;

(4)　Due to the small sample size in the pre-training stage, regularization should be considered in order to prevent over-fitting. The regularization form shown in Equation (4) is adopted, taking $\lambda = 20$.

To sum up, the empirical values for parameters are shown in Table 3.

**Table 3.** Empirical values of AE parameters.

| Name | Value |
| --- | --- |
| Activation function | Tanh |
| Learning machine | Adam |
| Learning rate | 0.01 |
| Regularization coefficient | 20 |
| Structure | 13-8-13 |

The tensorflow 2 AI framework based on Python is used in this paper. Under the given parameters, the loss function changes with epochs, as shown in Figure 9. It can be seen that after 500 epochs, the minimum loss (*loss\**) on the test set is 0.040.

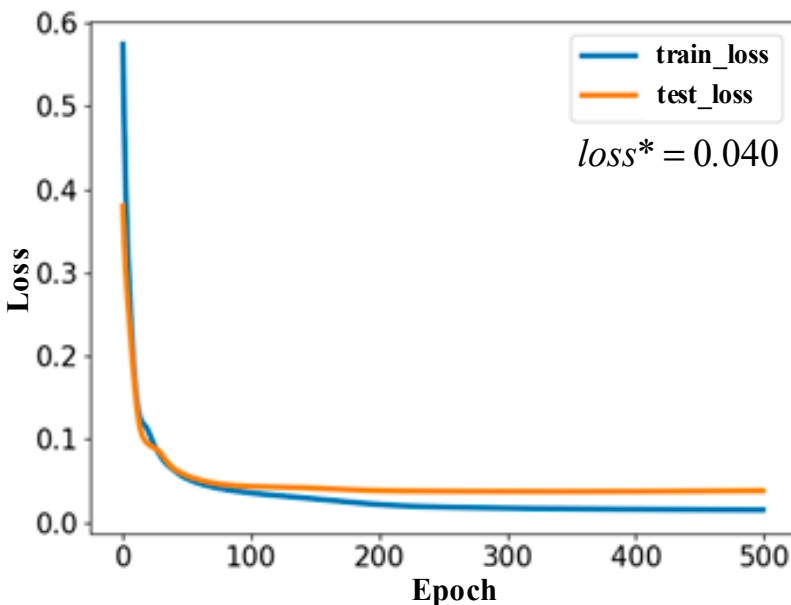

**Figure 9.** Iteration of AE loss.

In the stage of fine-tuning, the MLP network architecture is set to 13-16-32-16. As to the selection of activation function, the neural network regression problem generally chooses the ReLU activation function, which can alleviate the problem of gradient disappearance or explosion in deeper neural networks. Other parameters of the network are consistent with AE.

In order to make the whole network develop towards optimizing the AE network, the loss function can be modified, which is shown in Equation (23).

$$L = \eta_1 L_1 + \eta_2 L_2 \tag{23}$$

where $L_1$ represents the reconstruction loss of AE and $L_2$ represents the predicted loss of MLP. $\eta_1$ and $\eta_2$ are balance coefficients, used to balance the weight between reconstruction and prediction losses. The purpose of fine-tuning is to obtain a more refined low-dimensional representation of the sample, so the reconstruction loss should be fully considered. This paper takes $\eta_1 = 0.4$ and $\eta_2 = 0.6$.

Since the pre-training has adjusted the network weight to a reasonable range, the learning rate can be appropriately reduced at the training stage. In this paper, the initial learning rate is adjusted from 0.01 to 0.002, and the change of loss with epoch is shown in Figure 9. Due to the role of pre-training, AE shows low loss at the beginning of training. As the number of iterations increases, the loss firstly decreases and then begins to rise, which means that the model shows overfitting. After tuning, the optimal loss of AE is 0.037, which is less than the loss in Figure 10 (0.004). It can be seen that fine-tuning is effective.

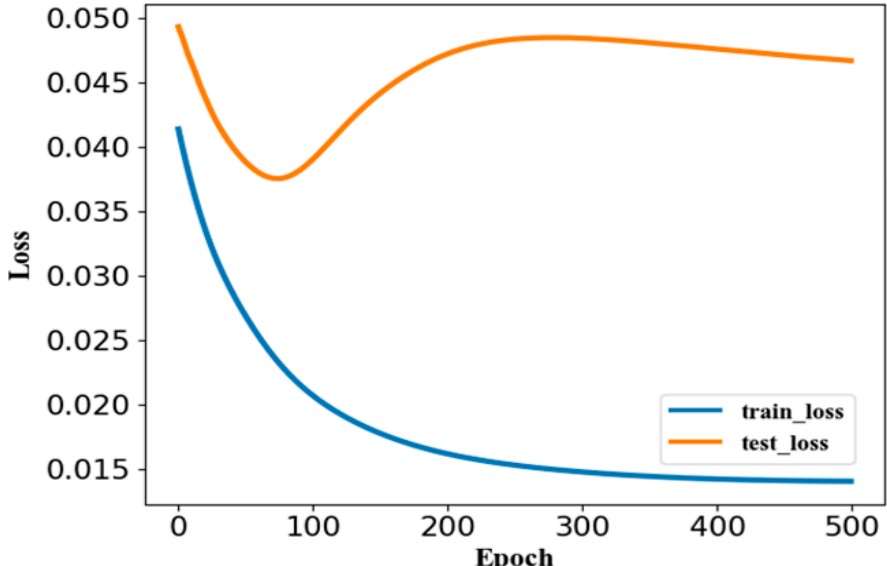

**Figure 10.** AE loss at the tuning stage.

Then the encoder of AE is extracted after training, with the weight retained. This part is used to reduce the representation of screw preloads and the results are shown in Table 4, where HD means hidden dimension.

**Table 4.** Representation reduction of screw preloads.

|  | HD 1 | HD 2 | . . . | HD 8 |
|---|---|---|---|---|
| Sample 1 | −0.6693 | 1.0457 | . . . | −0.4189 |
| Sample 2 | −0.6443 | 0.7543 | . . . | −0.5363 |
| . . . | . . . | . . . | . . . | . . . |
| Sample44 | −0.5054 | 1.1953 | . . . | −1.0646 |
| Sample45 | 0.0134 | 1.2093 | . . . | −0.4793 |

### 3.3. Online Prediction of Assembly Error

After the representation reduction of the dataset, the number of features is reduced from 13 to 8, and then the data is passed to KELM as input. For KELM, the kernel function needs to be determined first. Among the three kernel functions, the Polynomial kernel

function and Gaussian kernel function need the corresponding constant coefficient values, and these values to consider are shown in Table 5.

**Table 5.** Optional hyper-parameters of the kernel function.

| Name | Value |
|------|-------|
| $a$ | 1, 2, 3 |
| $b$ | 1, 2, 3 |
| $p$ | 1, 2, 3 |
| $\sigma$ | 10, 50, 100 |

Although three kinds of kernel functions are used here, it is easy to cause the problem of combinatorial explosion when considering hyper-parameters of the kernel function. In order to show the process and effectiveness of the method, this paper only tests 9 kinds of lifting orders, and selects the optimal lifting order from them.

Based on the optional constant values provided in Table 4 and 9 lifting sequences, the training set was first trained and then the prediction performance was tested on the test set. In order to further verify the effectiveness of representation reduction by AE, the experiment also compared the prediction result with the one without representation reduction. The final results are shown in Table 6, where L, G, and P in the table represent Linear kernel, Gaussian kernel, and Polynomial kernel, respectively.

**Table 6.** Optimal results under nine lifting sequences.

| Lifting Sequence | $a$ | $b$ | $p$ | $\sigma$ | MSE (Before Reduction) | MSE (After Reduction) |
|------------------|-----|-----|-----|----------|------------------------|-----------------------|
| L-G-P | 1 | 1 | 1 | 10 | 0.130062476 | 0.154568212 |
| L-P-G | 3 | 3 | 1 | 10 | 0.105822771 | 0.131452813 |
| G-L-P | 1 | 1 | 1 | 10 | 0.130104604 | 0.130115361 |
| **G-P-L** | **3** | **3** | **1** | **100** | **0.105821111** | 0.122455686 |
| P-G-L | 3 | 3 | 1 | 100 | 0.105821115 | 0.115254136 |
| P-L-G | 3 | 3 | 1 | 10 | 0.105823524 | 0.123654616 |
| G-G-G | - | - | - | 50 | 0.111207806 | 0.115623874 |
| L-L-L | - | - | - | - | 0.233036629 | 0.201251368 |
| P-P-P | 3 | 3 | 1 | - | 0.105893658 | 0.113456438 |

It can be seen from Table 6 that the performance of the nine lifting sequences is similar. Taking G-P-L as the lifting sequence can bring the smallest prediction error, and the corresponding optimal parameters are $a = 3$, $b = 3$, $p = 1$, $\sigma = 100$. In addition, by comparing the prediction results before and after representation reduction, it can be found that the prediction accuracy after reduction is slightly higher, which shows that the representation reduction of sample data is effective.

The above experimental data is only the prediction result of KELM under the boosting strategy. It is an offline data prediction to determine the parameters of Boosting-OSKELM. In order to verify the online prediction performance of Boosting-OSKELM, a simple sequential addition principle is adopted, as shown in Figure 11. The new sample points generated at a time $t + 1$ can be directly added to the training set, and incremental learning is carried out according to Equation (20). Finally, the error between the prediction and ground truth will be obtained.

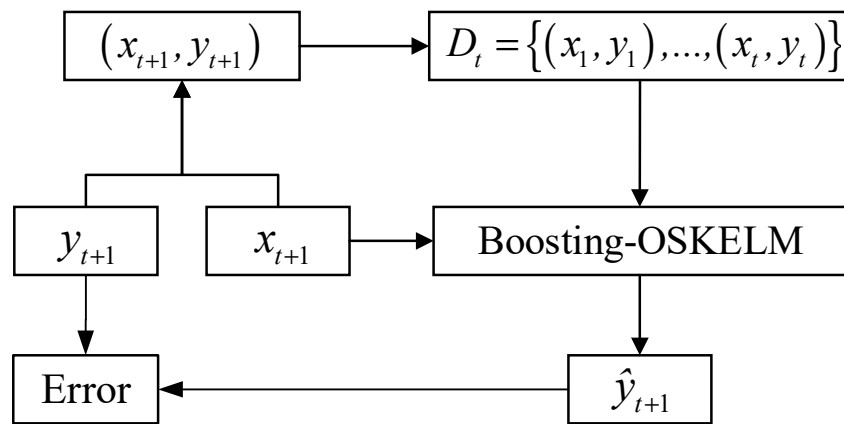

**Figure 11.** Sequential addition principle.

For further verifying the online learning ability of Boosting-OSKELM, the artificial neural network (ANN) is selected for comparison. Table 7 shows the comparisons of calculation efficiency between Boosting-OSKELM and ANN, while the time and iteration are the average value under 10 repeated experiments. As the former model relies on matrix transformation rather than gradient descent, the time consumed by which is within 1s (0.85 s), while the latter one accounts for nearly 7.2 s, with an early stop mechanism and converging at the 15th iteration. Also, the average MSE of Boosting-OSKELM and ANN are 0.061 and 0.122. Thus it can be seen that the proposed model is superior to traditional ANN at computing speed, which is more suitable in the field of online predicting. The online prediction results are shown in Figure 12. According to this figure, with the increase of samples, the prediction error shows a downward trend. From the perspective of the online prediction process, due to the randomness of sample distribution, there are some fluctuations in the online prediction process, but the fluctuations of Boosting-OSKELM is smaller than ANN, which shows that Boosting-OSKELM has stronger online learning adaptability.

**Table 7.** Average time consumption between two models under 10 repeated experiments.

| Model | Time (s) | Iteration |
|---|---|---|
| Boosting-OSKELM | 0.85 | / |
| ANN | 7.2 | 15 |

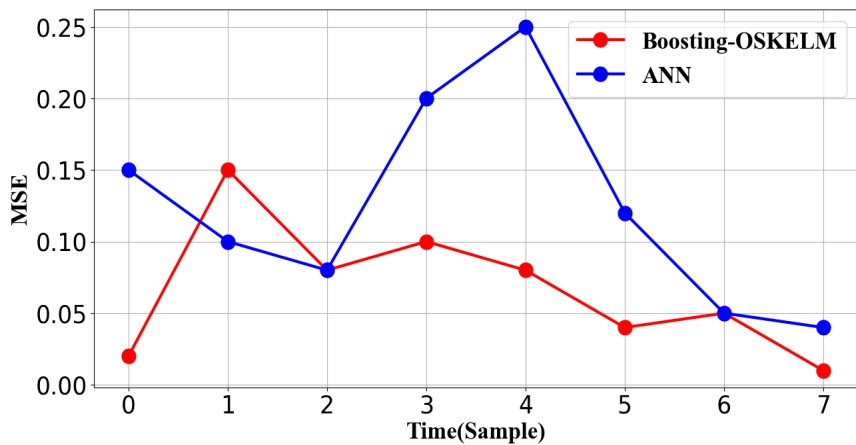

**Figure 12.** Online prediction performance over time.

## 4. Conclusions

As to the assembly of the array antenna, due to the production mode, the sample data of the array antenna presents the characteristics of high dimension and small samples, which brings difficulties to the prediction of its assembly accuracy. Therefore, this paper presents a data representation reduction method based on AE with fine-tuning trick and an online prediction method based on Boosting-OSKELM. The experiment results show that the average MSE of Boosting-OSKELM and ANN is 0.061 and 0.12, and the time consumption is 0.85 s and 15 s respectively. After analysis and discussion, the main conclusions are as follows.

(1) The representation reduction by AE can not only remove the redundant information in the original data but also meet the real-time requirements in the digital twin.
(2) With the help of multiple kernel functions and ensemble learning, Boosting-OSKELM can better adapt to online nonlinear learning problems. Compared with traditional ANN, its generalization performance is relatively stable. Therefore, the proposed method shows potential in other small sample problems.

As to the research in the future, although this paper has taken care of the small sample problem, more data needs to be sampled in order to improve the accuracy and robustness of the model. Also, the implementation of the presented model requires support from hardware and software. Therefore, the communication mode between intelligent prediction algorithms, parallel computing of industrial big data, and efficient storage can be the next research direction.

**Author Contributions:** Conceptualization, Y.T.; data curation, Y.T. and T.Z.; format analysis, Y.T.; investigation, M.W.; methodology, Y.T.; validation, M.W. and T.Z.; writing—original draft, Y.T.; writing—review and editing, M.W. and T.Z. All authors have read and agreed to the published version of the manuscript.

**Funding:** This work is supported by the National Key Research and Development Program of China (grant No. 2020YFB1710300, No. 2020YFB1710303) and the MOE (Ministry of Education in China) Project of Humanities and Social Sciences (No.17YJC630139).

**Institutional Review Board Statement:** Not applicable.

**Informed Consent Statement:** Not applicable.

**Data Availability Statement:** Data sharing not applicable.

**Conflicts of Interest:** The authors declare no conflict of interest.

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
