# Peer review of "Prediction of Array Antenna Assembly Accuracy Based on Auto-Encoder and Boosting-OSKELM"

_processes, doi:10.3390/pr11051460_

Round 1

Reviewer 1 Report

1.It would be helpful to provide more context about the specific challenges and limitations of current methods for predicting assembly accuracy of array antennas, and how the proposed method addresses these challenges. This can help readers better understand the motivation and contribution of the paper.

2.While the proposed method is shown to have strong robustness in prediction accuracy and online learning ability, it would be helpful to provide more details on the experimental setup and evaluation metrics used to demonstrate this. Additionally, it would be useful to compare the proposed method with other state-of-the-art methods for predicting assembly accuracy of array antennas, and discuss the strengths and limitations of each approach.

3.It would be helpful to provide more details on the implementation and computational complexity of the proposed method, and how it compares to other methods in terms of computational efficiency and scalability.

4.While the proposed method is specifically designed for predicting assembly accuracy of array antennas, it would be interesting to discuss the potential applications of the method in other domains where similar challenges of high dimensionality and small sample size exist.

5.It would be helpful to provide more information on the data used in the experiments, such as the size and characteristics of the dataset, and how it was collected and processed. This can help readers better understand the generalizability and applicability of the proposed method to real-world scenarios.

Reviewer 2 Report

The manuscript scientifically sounds good; however, I have the following major and minor concerns:

- Proofreading is essentially required.

- Paper organisation and presentation could be improved for better readability.

- Abstract and Conclusion require improvement based on the typical structure.

- Some figures (numbers, percentages, etc.) are required in the abstract.

- For the mentioned modern networking strategies and new infrastructure/environment, the authors must expand the literature by citing and adding the following references:

[1]     H. W. Oleiwi and H. Al-Raweshidy, “Cooperative SWIPT THz-NOMA / 6G Performance Analysis,” Electronics, Vol. 11, No. 6, pp. 873, March 2022.

[2]     H. W. Oleiwi, N. Saeed, and H. Al-Raweshidy, "Cooperative SWIPT-Hybrid-NOMA pairing scheme considering SIC im-perfection for THz communications," In Proceedings of IEEE 4th Global Power, Energy, and Communication Conference (GPECOM), Cappadocia, Turkey, 2022.

[3]     H. W. Oleiwi and H. Al-Raweshidy, “SWIPT-Pairing Mechanism for Channel-Aware Cooperative H-NOMA in 6G Terahertz Communications,” Sensors, Vol. 22, No. 16, pp. 6200, Aug. 2022.

- For the mentioned AI-based (Auto-Encoder, Boosting-OSKELM, Accuracy, Complexity, Performance, Evaluation, etc.) processes, the authors must expand the strategies by citing and adding the following references:

[4]     H. W. Oleiwi, D. N. Mhawi, and H. Al-Raweshidy, "MLTs-ADCNs: Machine Learning Techniques for Anomaly Detection in Communication Networks," IEEE Access, Vol. 10, pp. 91006–91017, Aug. 2022.

[5]     H. W. Oleiwi, D. N. Mhawi, and H. Al-Raweshidy, "A Meta-Model to Predict and Detect Malicious Activities in 6G-Structured Wireless Communication Networks," Electronics, Vol. 12, No. 3, pp. 643, Jan. 2023.

- The authors need to write the introduction section according to the comments above, expanding it by adding the explained issues in the references above.

- Please avoid long sentences and informal expressions.

- It is highly recommended to restructure the paper introduction using Introduction, Background, and Related Work sections. And also, a section for the implementation environment and setup.

- It is (optionally) preferred if you could write a table for related works comparisons to emphasize the outperformance of their work over the state of the art.

- Re-draw Figure 7, please, resizing the legends' font size, and do the same (with the best quality) for other figures if applicable.

- Split up methodology, implementation environment (setup and datasets used), and results/discussion separately in different clear sections.

- Replace the old references should you have any alternatives.

Reviewer 3 Report

General comments

After the introduction where author presented the need for a method which predicts efficiently and quickly the accuracy of an assembly process, he explained the traditional idea of the Online Learning Kernel Sequential Extreme Learning Machine (OSKLEM) as reported also in the literature [1, 2]. Author, then, explained in detail his Boosting-OSKLEM algorithm improvement “based on the idea of boosting, in order to squeeze the performance of the learner as much as possible” as he suggested in section 2. In section three, author presented a case study related to antenna array assembly process with results after introducing the concept of “fine tuning” to shorten the number of assembly cycle. At the end, and for confirming efficiency of his method, author presents a comparison between Boosting-OSKELM and Artificial Neuron Network (ANN).

I found this paper very interested, its English is very nice, the sequence of ideas is clear, its organization is exact and its topic matches perfectly to the Journal’ aim and scope

In the view of the above, I think this work deserves worthy to be published at his actual form

Remark: Please verify if this version has not been published before because I found this preprint version on “Research Square” (https://www.researchsquare.com/article/rs-1400540/v1):

“Miao Wang, Yifei Tong, Tong Zhou , “Prediction of Array Antenna Assembly Accuracy Based on Auto-Encoder and Boosting-OSKELM”, 22 march 2022, https://doi.org/10.21203/rs.3.rs-1400540/v1”

References of reviewer

1.       1. S. Scardapane, D. Comminiello, M. Scarpiniti and A. Uncini, "Online Sequential Extreme Learning Machine With Kernels," in IEEE Transactions on Neural Networks and Learning Systems, vol. 26, no. 9, pp. 2214-2220, Sept. 2015, doi: 10.1109/TNNLS.2014.2382094

2.       2. Wan-Yu Deng, Yew-Soon Ong, Puay Siew Tan, Qing-Hua Zheng, Online sequential reduced kernel extreme learning machine, Neurocomputing, Volume 174, Part A, 2016, Pages 72-84, ISSN 0925-2312, https://doi.org/10.1016/j.neucom.2015.06.087

Round 2

Reviewer 2 Report

Well revised